# Adequacy and Vitamin D in the Preoperative Period of Roux-en-Y Gastric Bypass, Bariatric Surgery, Can Protect Metabolic Health in Metabolically Healthy and Unhealthy Individuals

**DOI:** 10.3390/nu14030402

**Published:** 2022-01-18

**Authors:** Suelem Pereira da Cruz, Sabrina Cruz, Silvia Pereira, Carlos Saboya, Juliana Castelar Lack Veiga, Andréa Ramalho

**Affiliations:** 1Center for Research on Micronutrients, Federal University of Rio de Janeiro (NPqM/UFRJ), Rio de Janeiro 21941-902, Brazil; sabrina.cruz.ufrj@gmail.com (S.C.); aramalho.rj@gmail.com (A.R.); 2Multidisciplinary Center for Bariatric and Metabolic Surgery, Federal University of Rio de Janeiro (NPqM/UFRJ), Rio de Janeiro 21941-902, Brazil; se.pereira@gmail.com; 3Brazilian Society of Bariatric and Metabolic Surgery, Rio de Janeiro 22280-020, Brazil; cjsaboya@carlossaboya.com.br; 4Center for Research on Micronutrients, Federal University of Fluminense, Rio de Janeiro 24220-008, Brazil; julianacastelar@id.uff.br

**Keywords:** bariatric surgery, obesity, vitamin D

## Abstract

Evaluating the influence of vitamin D concentrations together with preoperative metabolic phenotypes on remission of chronic noncommunicable diseases (CNCDs) after 6 months of Roux-en-Y gastric bypass (RYGB). Cross-sectional analytical study comprising 30 adult individuals who were assessed preoperatively (T0) and 6 months (T1) after undergoing RYGB. Participants were distributed preoperatively into metabolically healthy obese (MHO) and metabolically unhealthy obese (MUHO) individuals according to HOMA-IR classification and to the adequacy and inadequacy of vitamin D concentrations in the form of 25(OH)D. All participants were assessed for anthropometric characteristics, biochemical variables, and presence of CNCDs. The statistical program used was the SPSS version 21. In face of vitamin D adequacy and regardless of the metabolic phenotype classification in the preoperative period, the means found for HOMA-IR allowed us to define them as metabolically healthy 6 months after RYGB. Only those with vitamin D inadequacy with the MUHO phenotype showed better results regarding the reduction of glucose that accompanied the shift in serum 25(OH)D concentrations from deficient to insufficient. It is possible that preoperative vitamin D adequacy, even in the presence of an unhealthy phenotype, may contribute to the reduction of dyslipidemia and improvement in cholesterol. It is suggested that preoperative vitamin D adequacy in both phenotypes may have a protective effect on metabolic health.

## 1. Introduction

Bariatric surgery aims to reduce the total body mass with consequent remission of chronic noncommunicable diseases (CNCDs), which can happen more intensely in the minimum period of 6 months after the procedure [1]. Several factors may be involved in this remission, among which we highlight the metabolic phenotypes and serum concentrations of vitamin D.

The scientific literature, when comparing metabolically healthy obese (MHO) or unhealthy obese (MUHO) individuals, shows that MUHO presents characteristics that can facilitate the onset of obesity-associated diseases, such as inadequate functionality of adipose tissue that, in turn, favors the appearance of lipotoxic products, presence of insulin resistance (IR), greater ectopic and visceral fat storage that contribute to the development of cardiometabolic diseases [2,3,4,5].

The weight loss process promoted by bariatric surgery may be related to lipolytic gene expression that correlates inversely with biomarkers of lipid and glucose metabolism [6]. However, some studies evaluating the preoperative interference of the MHO and MUHO phenotypes on remission of CNCDs after a period of 6 to 24 months after bariatric surgery found contradictory results [7,8,9]. Although weight loss resulting from bariatric surgery may promote metabolic benefits, there may be disorders originating from the procedure. Such disorders may contribute to the development of nutritional deficiencies such as vitamin D deficiency (VDD) [10,11,12]. This scenario may be even more worrisome due to (1) a frequent decrease in serum concentrations of this nutrient previous to surgery, (2) the strongly established relationship of this vitamin with body fat distribution, and (3) its peculiar role in regulating several metabolic processes [13,14].

However, to date, there are no studies investigating the relationship between vitamin D and CNCD considering the metabolic phenotypes after bariatric surgery. Thus, the aim of the present study is to evaluate the influence of vitamin D concentrations together with the preoperative metabolic phenotypes on remission of CNCDs 6 months after Roux-en-Y Gastric Bypass (RYGB).

## 2. Materials and Methods

This is a cross-sectional study of the analytical type consisting of 30 adult individuals with obesity who underwent RYGB, selected by convenience at the Multidisciplinary Center for Bariatric and Metabolic Surgery located in the city of Rio de Janeiro, Brazil, in the period from July 2019 to September 2020.

Participants were evaluated preoperatively (T0) in relation to the inclusion and exclusion criteria established for the research and followed up for 6 months (T1) after undergoing RYGB. They were further distributed preoperatively into MHO (metabolically health obese) and MUHO (metabolic unhealthy obese) according to HOMA-IR classification, as well as to the adequacy and inadequacy of vitamin D concentrations (25(OH)D) according to the criteria of Holick et al. (2012) [15,16,17]. Inclusion criteria were adults of both sexes classified according to body mass index (BMI) ≥ 35 kg/m^2^ and age ≥ 20 and <60 years preoperatively, who formally authorized their enrollment by signing the informed consent form. Exclusion criteria were: Previous disabsorptive and restrictive surgeries, intestinal disabsorptive syndromes, neoplasms, use of drugs for total body mass loss, alcohol consumption higher than 20 g/day for women and 40 g/day for men, pregnant or nursing women, kidney insufficiency and liver diseases, except nonalcoholic fatty liver disease, endocrinopathies (hypothyroidism, hypercortisolemia), acute and chronic infections, use of anticonvulsant drugs or drugs that interfere with vitamin D metabolism, use of medication or supplement rich in vitamin D for 2 months prior to the first laboratory test. The instrument used for data collection was previously tested and consisted of a form filled out by a single interviewer through consultation of medical records. All the members of this study were evaluated for anthropometric variables and laboratory tests. Regarding the anthropometric variables, the total body mass (kg) and height (m) were measured to calculate the BMI in which the cutoff point adopted for class II and III obesity classification was ≥35 kg/m^2^, according to the WHO recommendation [18]. Biochemical tests were performed to evaluate the lipid profile by the Enzymatic Colorimetric method, measuring total cholesterol, high density lipoprotein cholesterol (HDL-c), low-density lipoprotein cholesterol (LDL-c), triglycerides; fasting glycemia by the enzymatic method, glycosylated hemoglobin (HbA1c) by turbidimetry, basal insulin by chemiluminescence, evaluation of the homeostatic model-insulin resistance by specific calculation of the Homeostasis Model Assessment Estimate for Insulin Resistance (HOMA-IR). The latter was used to assess insulin resistance, and values equal to or above 2.5 were used as cutoff point [16,17]. The study was approved by the Research Ethics Committee of the Hospital Universitario Clementino Fraga Filho and the Faculty of Medicine of the Federal University of Rio de Janeiro, under n° 360/60, Research Protocol n° 011/10, on 15 May 2010.

Vitamin D analysis was performed in the form of 25(OH)D and 1,25(OH)2D by means of High Efficiency Liquid Chromatography with an ultraviolet detector (HPLC-UV) [19]. Thus, this nutrient was evaluated in the form of 25(OH)D and classified as showing deficiency (≤20 ng/mL) or insufficiency (≥21 ng/mL lower or equal to 29 ng/mL). Participants were classified as inadequate when they showed insufficiency and deficiency, whereas those with values ≥30ng/mL lower or equal to 100 ng/mL were classified as adequate [15].

The overall technical requirements for adequately obtaining blood pressure (BP), as well as the definition of the cutoff point equal to or greater than 140/90 mmHg, which is considered hypertensive, followed the specifications of the VI Brazilian Guidelines on Hypertension [20].

For diagnosing type II diabetes mellitus (2DM), the American Diabetes Association criterion was applied [16]. Thus, 2DM was diagnosed when fasting serum glucose was ≥ 126 mg/dL, or capillary blood glucose with symptoms of hyperglycemia was equal to or greater than 200 mg/dL, or 2 h after 75 g of glucose overload with values equal to or greater than 200 mg/dL or glycated hemoglobin equal to or greater than or 6.5. To evaluate the presence of dyslipidemias, the cutoff points of the V Brazilian Guidelines on Dyslipidemias and Prevention of Atherosclerosis were used [17]. The cut-off points were added to the tables.

Statistical analyses were performed using the Statistical Package for the Social Sciences (SPSS) for Windows version 21.0. The Kolmogorov–Smirnov test was performed to evaluate the normality of the sample. Measures of central tendency and dispersion (mean and standard deviation) were calculated for quantitative variables, and the Mann–Whitney or Kruskal–Wallis tests were used to compare the means. To test the homogeneity of proportions between categorical variables, the Pearson chi-square test, the Fisher’s exact test, and Linear Correlation (Spearman test) were applied between the non-parametric variables. The significance level adopted was 5% (*p* < 0.05).

## 3. Results

The study comprised 30 patients who previously underwent RYGB. Their evaluation was conducted before and 6 months after this surgical procedure. In the total sample, 60% were female and 40% male. To perform the evaluations, the patients were divided into groups, considering the metabolic phenotypes together with the adequacy/inadequacy of serum vitamin D concentrations, and were also matched for BMI, age and sex.

The mean HOMA-IR values, 6 months after RYGB, allow including in the healthy phenotype the individuals who had vitamin D adequacy preoperatively, independent of the metabolic phenotype classification (Table 1). Furthermore, there was a reduction in the BMI percentage independent of the subdivision of the groups. When the groups were only evaluated according to the preoperative phenotypes, we observed that 46.7% of the sample had a healthy metabolic phenotype and 53.3% had an unhealthy metabolic phenotype, and the means of the preoperative serum 25(OH)D concentrations of the total sample were 23.31 ± 9.62. Furthermore, 81.2% of those with MUHO transitioned to MHO 6 months after bariatric surgery. Additionally, both metabolic phenotypes had 25(OH)D means that allow their classification as insufficient in both segments (MHO: 27.67 ± 10.66 and MUHO: 21.36 ± 7.78; *p* = 0.151). Moreover, they presented similar means for almost all biochemical variables, except for 1,25(OH)2D, which was higher in the MUHO (MHO: 35.18 ± 8.53 and MUHO individuals: 47.40 ± 9.31; *p* = 0.003) and showed a strong correlation with HOMA-IR (0.982 *p* = 0.000). Nonetheless, such an increase in postoperative serum 1,25(OH)2D concentrations occurred only in the MUHO segment with preoperative 25(OH)D adequacy (MUHO with adequacy: 33.32 ± 1.14 and MUHO with inadequacy: 17.37 ± 3.37; *p* = 0.000).

When the groups were only analyzed according to the preoperative adequacy/inadequacy of 25(OH)D, the means in the segment of adequacy of this nutrient were 37.32 ± 6.44 and in those with inadequacy were 19.57 ± 5.09 (*p* = 0.000). Patients with preoperative 25(OH)D ≥ 30 ng/mL maintained their means after surgery, and also had reduced HOMA-IR means, which allows classifying individuals as MHO (T0:3.87 ± 3.40 and T1:1.25 ± 0.57; *p* = 0.007).

### 3.1. 25(OH)D ≥ 30 ng/mL and Metabolic Phenotypes

In relation to the patients who had an adequacy of 25(OH)D with the MHO phenotype, there was a reduction in glycated hemoglobin and insulin in which the latter variable also had a strong negative correlation with the metabolic phenotype at T0 (T0: r = −0.980; *p* = 0.020). In addition, 1,25(OH)2D increased after bariatric surgery (Table 1).

The segment of participants who had preoperative 25(OH)D adequacy together with MUHO had reduced mean and percentage of cholesterol inadequacy after surgery (Table 1), which in turn was strongly and negatively correlated with the metabolic phenotype (T0: −r = 0.955; *p* = 0.045). Moreover, in this segment, 25(OH)D was strongly and negatively correlated with BMI (T0: r = −0.997; *p* = 0.003), insulin (T0: r = −0.975; *p* = 0.025) and positively correlated with 25(OH)D (T1: r = 0.977; *p* = 0.023) 6 months after RYGB. Furthermore, the presence of dyslipidemia was reduced, and there were no cases of MUHO after 6 months (Table 1).

### 3.2. 25(OH)D < 30 ng/mL and Metabolic Phenotypes

When there was preoperative inadequacy of 25(OH)D with the MHO phenotype, a reduction in insulin, glycated hemoglobin, cholesterol, and LDL was observed postoperatively. Lower percentages of cholesterol inadequacy cases were also observed (Table 2). Insulin showed strong positive correlation with the metabolic phenotypes (T0: r = 0.820; *p* = 0.004) and negative correlation with 25(OH)D (T0: r = −0.747; *p* = 0.013), both assessed preoperatively. This nutrient also correlated negatively with glucose (T0: r = −0.693; *p* = 0.026) and positively with HDL (T0: r = 0.665; *p* = 0.036) at T0, as well as with postoperative 25(OH)D (T1: r = −0.747; *p* = 0.013).

In the segment of vitamin D inadequacy and MUHO, there was a reduction of insulin, glycated hemoglobin, cholesterol, LDL, glucose and increase in 25(OH)D shifting from deficient to insufficient, as well as a decrease in the percentage of inadequate insulin, cholesterol and MUHO cases (Table 2). Moreover, postoperatively, a positive correlation was found between the metabolic phenotypes assessed at T0 with VLDL (T1: r = 0.603; *p* = 0.038) and HOMA-IR (T1: r = 0.607; *p* = 0.036). Regarding preoperative 25(OH)D, there was a negative correlation with HOMA-BETA (T0: −r = 0.659; *p* = 0.020) at T0 and TG (T0: r = −0.696; *p* = 0.012 and T1: r = −0.610; *p* = 0.035) at both times.

## 4. Discussion

It is estimated that the inadequacy of 25(OH)D in the pre- and postoperative period is a recurring event and may reach up to 80% of this segment [21,22,23]. This nutrient, in turn, may be related to metabolic phenotypes in obesity, considering that individuals of the MHO phenotype may have higher concentrations of 25(OH)D, possibly justified by a healthier metabolic profile compared to individuals of the MUHO phenotype [24]. Furthermore, it has been suggested that 25(OH)D may offer a protective metabolic effect in the MHO group [25]. It has also been reported that VDD may be related to cardiovascular events, especially in the presence of metabolic alterations associated with obesity [26,27].

In this sense, some studies argue about the possibility of 25(OH)D being considered a clinical marker for a healthy metabolic profile, as well as whether this nutrient can predict the progression from the MHO status to the MUHO, considering that VDD may contribute to the increased risk of developing CNCDs [28,29,30,31,32,33,34]. Inclusively, VDR expression may underlie the various effects of 25(OH)D and provide a mechanistic basis for the association between VDD and diseases that are related to obesity [34,35,36], in which it has been suggested that the low-grade inflammation in obesity may be linked to VDD [37].

Thus, in the context of bariatric surgery, the presence of a healthy phenotype and preoperative adequacy of 25(OH)D could be related to more expressive metabolic benefits after bariatric surgery, however, there are no studies that address this issue. In this aspect, the present study found that when there is 25(OH)D adequacy, the means found for HOMA-IR, 6 months after RYGB, allow defining these individuals as metabolically healthy, regardless of the preoperative classification of the metabolic phenotype. Therefore, it is possible that preoperative 25(OH)D adequacy in both phenotypes may have a protective effect on metabolic health.

Although no significant differences were found between the phenotypes regarding serum 25(OH)D concentrations after bariatric surgery, MUHO participants had higher mean 1,25(OH)2D, which, in turn, showed a strong correlation with HOMA-IR. However, this post-surgical increase occurred only in the MUHO segment with vitamin D adequacy (25(OH)D). Thus, it is suggested that the increased 1,25(OH)2D concentrations in MUHO with 25(OH)D adequacy found in the present study may indicate an increased need for it, which in turn may have contributed to metabolic protection of these patients after surgery. Even in this segment, there were no cases of MUHO 6 months after RYGB. Thus, it is possible that calcitriol may attenuate the action of preadipocytes in the basal release of IL-8 and IL-6 and in the action of macrophages, leading to a reduction in the production of MCP-1, which ratifies the anti-inflammatory effect, facilitating reduction of MUHO [38,39,40]. Moreover, some studies have reported the presence of vitamin D receptors (VDR) in ventricular cardiomyocytes and fibroblasts, suggesting an important action of this nutrient in the maintenance of cardiometabolic health [41,42]. In segments of patients with hypertension, 2DM and obesity, foam cell formation in isolated macrophages can be suppressed by 1,25(OH)2D and the mechanism of action involves the reduction of LDL uptake [43]. In addition, experimental study suggests that the progression of coronary artery disease may be accelerated by the presence of VDD, due to the increased activation of the nuclear factor-kB, a nutrient indirectly related to the anti-inflammatory function [44].

In the context of type 2 diabetes mellitus pathophysiology, it has been reported that vitamin D may exhibit some actions in pancreatic beta cells such as activation of the VDR; and binding of 1,25(OH)2D to the VDR, which promotes transcription of regulated genes to its active form. In addition, it is also responsible for the presence of the vitamin D responsive element (VDRE) in the promoter gene of insulin and the activation of the human insulin gene transcription caused by 1,25(OH)2D [36]. According to experimental and epidemiological review studies, there was an association of VDRE with presence of IR, 2DM and decreased insulin secretion. Nevertheless, meta-analysis of clinical trial shows no significant effect of vitamin D supplementation on glycemic control [45]. In the present study, a decrease in mean glycated hemoglobin and insulin was found in all groups. Furthermore, there was no presence of 2DM after RYGB. Thus, it is possible that the reduction of body weight and, therefore, reduction of the inflammatory process promoted by obesity, may be associated with the improvement of these variables. Moreover, only patients with preoperative inadequate vitamin D with MUHO showed better results regarding the reduction of glucose that accompanied the increase on serum concentrations of 25(OH)D, which shifted from deficient to insufficient. Additionally, 75% transitioned to the healthy phenotype. Thus, it is suggested that this increase in vitamin D in the post-surgical period may contribute not only to a higher percentage of the MHO cases, but also to glucose reduction. In this sense, it has been reported that this vitamin has the ability to improve glucose homeostasis by optimizing the function of pancreatic β cells, thus increasing insulin secretion [46]. Moreover, it may also be associated with glucose uptake in adipocytes [37], a fact that may justify the present result.

Regarding the variables related to dyslipidemia, a reduction in cholesterol was observed. Besides, in the case of patients classified with vitamin D adequacy together with MUHO, there was also a reduction in dyslipidemia. In this sense, the literature indicates that vitamin D can reduce the accumulation of cholesterol in macrophages, as well as the uptake of LDL in atheroma plaques [47]. Furthermore, through the modulation of inflammatory responses, with reduce expression of TNF-α, IL-6, IL-1, and IL-8 of monocytes, f TNF-α, IL-6, IL-1, and IL-8 of monocytes, Vitamina D may influence the pathophysiology of atherosclerosis [48,49]. In addition, it modulates thrombomodulin expression in monocytes, affecting platelet aggregation and thrombogenic activity [50]. However, there are no studies associating dyslipidemia with metabolic phenotypes. Therefore, the present study is the first to find this association. Thus, due to the widely reported metabolic actions of vitamin D, the adequacy of this vitamin in the preoperative period, even in the presence of an unhealthy phenotype, may contribute to the reduction of dyslipidemia and improvement of cholesterol concentrations. Thus, the present study suggests that because of these possible actions of vitamin D, even in the presence of an unhealthy phenotype, preoperative vitamin D adequacy may contribute to dyslipidemia reduction and cholesterol improvement.

It is worth mentioning that this increase of vitamin D may present pleiotropic effects on cardiovascular health, such as nuclear receptor activation in cardiomyocytes, vascular endothelial cells, and regulation of the renin-angiotensin-aldosterone system. Thus, its deficiency in humans is associated with vascular dysfunction; arterial stiffness; left ventricular hypertrophy, SAH, and dyslipidemia [51,52,53,54]. However, there were no changes in the cases of postoperative SAH.

The present study has the limitation of sample size, however, nonparametric statistical analyses minimize this feature. Furthermore, this is the first study to assess, in a joint manner, the inherent results of vitamin D adequacy together with metabolic phenotypes on remission of CNCDs after RYGB. This fact may contribute to strategies to be implemented preoperatively to minimize the presence of CNCDs in the early postoperative period.

## 5. Conclusions

Given the results, it was possible to observe that bariatric surgery can negatively influence the achievement of better metabolic profiles, including changes in serum 25(OH)D. These changes were more prominent in those with vitamin D adequacy in the preoperative period since it contributed to dyslipidemia reduction and cholesterol improvement after surgery in both metabolic phenotypes. Additionally, the postoperative increase in 25(OH)D was accompanied by a reduction in glucose in those with vitamin D inadequacy and presence of the MUHO phenotype. Therefore, it is suggested that further research on this topic should be conducted to identify whether adequate preoperative vitamin D in both metabolic phenotypes may contribute to a protective metabolic effect.

## Figures and Tables

**Table 1 nutrients-14-00402-t001:** Means or percentage of inadequacy of body and biochemical variables and presence of chronic noncommunicable diseases considering 25(OH)D ≥ 30 ng/mL together with different metabolic phenotypes before and after Roux-en-Y Gastric Bypass bariatric surgery.

	25(OH)D ≥ 30 ng/mL and MHO T0	25(OH)D ≥ 30 ng/mL and MHO T1	*p*-Value	25(OH)D ≥ 30 ng/mL and MUHO T0	25(OH)D ≥ 30 ng/mL and MUHO T1	*p*-Value
Body Variables Means/Inadequacy % and reference values						
Age	44.50 ± 2.88	44.50 ± 2.88	1.000	38.00 ± 13.85	38.00 ± 13.85	1.000
Weight	135.40 ± 28.29	102.15 ± 23.99	0.245	126.25 ± 2.75	94.65 ± 12.90	0.021 *
BMI (kg/m^2^)	44.49 ± 2.29	33.51 ± 3.61	0.021 *	45.88 ± 2.75	34.26 ± 2.14	0.021 *
BMI(Obesity)	100%	75%	0.046 *	100%	100%	0.018 *
Biochemical Variables						
Means and reference values						
HOMA IR	1.71 ± 0.21	1.14 ± 0.48	0.110	6.03 ± 3.80	1.37 ± 0.71	0.021 *
HOMA BETA	101.15 ± 24.31	77.95 ± 24.74	0.248	239.67 ± 229.40	172.97 ± 159.15	0.149
Insulin (2,0 a 17 mcU/mL)	7.62 ± 1.10	2.17 ± 0.81	0.021 *	25.54 ± 15.66	8.30 ± 7.24	0.149
Blood Glucose (70–99 mg/dL)	87.75 ± 9.91	79.25 ± 17.83	0.564	98.50 ± 13.86	93.25 ± 16.15	0.248
Glycated Hemoglobin(<6.5)	5.15 ± 0.51	2.30 ± 0.64	0.020 *	5.40 ± 0.24	3.67 ± 1.10	0.058
Cholesterol(<100 mg/dL)	181.25 ± 64.95	104.07 ± 36.90	0.083	197.50 ± 48.47	120.25 ± 35.52	0.043 *
LDL(<100 mg/dL)	107.75 ± 45.77	76.00 ± 9.89	0.083	91.25 ± 3.30	82.00 ± 13.44	0.248
HDL [>40 mg/dL ( 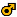 ); >50 mg/dL ( 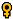 )]	41.25 ± 9.42	43.75 ± 24.85	0.468	40.75 ± 9.74	45.50 ± 7.76	0.309
VLDL (mg/dL)	25.75 ± 15.84	32.00 ± 20.01	0.773	24.25 ± 15.37	14.25 ± 4.34	0.243
TG (<150 mg/dL)	109.50 ± 43.74	90.75 ± 27.93	0.245	143.00 ± 64.99	135.75 ± 73.62	0.773
25(OH)D (ng/mL)	41.32 ± 7.27	45.10 ± 8.62	0.248	33.32 ± 1.14	27.70 ± 6.60	0.468
1,25(OH)2D (ng/mL)	37.47 ± 1.41	50.34 ± 12.89	0.021 *	52.75 ± 11.29	52.75 ± 9.14	0.663
Percentage of Inadequacy						
Glucose (mg/dL)	25%	0%	0.285	50%	50%	1.00
Glycated Hemoglobin	0.0%	0%	-	0.0%	0%	-
Insulin	0.0%	25%	0.285	16.7%	0%	0.102
Cholesterol (mg/dL)	50%	25%	0.465	100%	25%	0.028 *
LDL (mg/dL)	25%	0%	0.285	0.0%	0%	-
HDL (mg/dL)	75%	50%	0.465	75%	50%	0.465
TG (mg/dL)	25%	0%	0.285	25%	25%	1.00
Vitamin D [25(OH)D]	0.0%	0%	-	0.0%	0%	-
Chronic Noncommunicable Diseases						
Dislipidemia	75%	50%	0.465	100%	25%	0.028 *
Diabetes Mellitus	0.0%	0%	-	0.0%	0%	-
SAH	25%	25%	1.00	25%	25%	1.00
Unhealthy phenotype	0.0%	0%	-	100%	0%	0.005 *

The Mann–Whitney test was used for continuous variables and the Chi-square test for categorical variables (* *p* ≤ 0.05). Biochemical variables: HDL-c—high-density lipoprotein cholesterol; HOMA-IR—homoeostasis model assessment for insulin resistance; LDL-c—low-density lipoprotein cholesterol; VLDL: very low-density lipoprotein cholesterol; TG: triglycerides; 25(OH)D-25-hydroxyvitamin D and 1,25(OH)2D—1,25 dihydroxyvitamin D. Body variables: BMI—body mass index; Adequate: adequate; Insufficient: insufficient; Def: deficient. SAH: systemic arterial hypertension.

**Table 2 nutrients-14-00402-t002:** Mean or percentage of inadequacy of body and biochemical variables, and presence of chronic noncommunicable diseases considering 25(OH)D < 30 ng/mL together with different metabolic phenotypes before and after Roux-en-Y Gastric Bypass bariatric surgery.

	25(OH)D < 30 ng/mL and MHO T0	25(OH)D < 30 ng/mL and MHO T1	*p*-Value	25(OH)D < 30 ng/mL and MUHO T0	25(OH)D < 30 ng/mL and MUHO T1	*p*-Value
Body Variables Means of Inadequacy and reference values						
Age	48.80 ± 8.28	48.80 ± 8.28	1.00	46.00 ± 10.21	46.00 ± 10.21	1.00
Weight	123.24 ± 15.33	103.80 ± 91.54	0.001 *	110.50 ± 15.15	79.91 ± 10.95	0.000 *
BMI (kg/m^2^)	40.18 ± 4.76	38.59 ± 29.26	0.001 *	39.97 ± 2.76	29.05 ± 3.62	0.000 *
BMI (Obesity)	0.0%	20%	0.136	100%	25%	0.000 *
Biochemical Variables						
Means and reference values						
HOMA IR	1.60 ± 0.57	4.63 ± 10.01	0.646	4.71 ± 1.52	6.65 ± 18.71	0.000 *
HOMA BETA	81.45 ± 40.76	68.15 ± 28.51	0.880	196.86 ± 137.24	136.40 ± 112.96	0.078
Insulin(2,0 a 17 mcU/ml)	8.49 ± 3.42	4.21 ± 2.77	0.007 *	22.05 ± 4.28	6.30 ± 5.36	0.000 *
Blood Glucose (70–99 mg/dL)	107.40 ± 34.37	91.90 ± 13.11	0.185	107.16 ± 26.67	88.75 ± 10.66	0.022 *
Glycated Hemoglobin(<6.5)	5.48 ± 0.81	3.74 ± 1.27	0.005 *	5.76 ± 0.81	3.61 ± 1.42	0.001 *
Cholesterol(<150 mg/dL)	197.80 ± 70.47	119.90 ± 30.64	0.003 *	231.91 ± 52.19	128.93 ± 44.67	0.000 *
LDL (<100 mg/dL)	103.90 ± 37.81	92.10 ± 8.15	0.449	149.50 ± 49.99	105.41 ± 24.46	0.043 *
HDL [>40 mg/dL ( 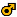 ); >50 mg/dL ( 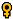 )]	46.50 ± 13.81	54.20 ± 10.66	0.212	42.58 ± 11.33	47.06 ± 11.11	0.285
VLDL (mg/dL)	23.90 ± 13.64	28.20 ± 27.18	0.820	28.61 ± 16.24	31.00 ± 36.64	0.402
TG (<150 mg/dL)	243 ± 448.71	84.50 ± 23.23	0.174	236.25 ± 322.95	141.08 ± 139.84	0.073
25(OH)D (ng/mL)	22.22 ± 5.53	27.70 ± 6.60	0.07	17.37 ± 3.59	25.26 ± 6.89	0.001 *
1,25(OH)2D (ng/mL)	34.27 ± 10.06	38.94 ± 13.36	0.426	45.62 ± 8.34	47.12 ± 9.20	0.644
Percentage of Inadequacy						
Blood Glucose (mg/dL)	40%	20%	0.329	50%	8.3%	0.059
Glycated Hemoglobin	25%	0%	0.305	75%	0%	0.064
Insulin	0.0%	10%	0.305	83.3%	16.7%	0.001 *
Cholesterol (mg/dL)	80%	30%	0.025 *	83.3%	41.7%	0.035 *
LDL (mg/dL)	50%	20%	0.160	66.7%	50%	0.408
HDL (mg/dL)	60%	20%	0.068	66.7%	50%	0.408
TG (mg/dL)	30%	0%	0.060	41.7%	25%	0.386
Vitamin D [25(OH)D]	100%	70%	0.060	100%	75%	0.064
Chronic Noncommunicable Diseases						
Dislipidemia	80%	40%	0.068	91.7%	66.7%	0.132
Diabetes	20%	0%	0.136	25%	0%	0.064
SAH	50%	50%	1.00	25%	25%	1.00
Unhealthy phenotype	0.0%	20%	0.136	100%	25%	0.000 *

The Mann–Whitney test was used for continuous variables and the Chi-square test for categorical variables (* *p* ≤ 0.05). Biochemical variables: HDL-c—high-density lipoprotein cholesterol; HOMA-IR—homoeostasis model assessment for insulin resistance; LDL-c—low-density lipoprotein cholesterol; VLDL: very low-density lipoprotein cholesterol; TG: triglycerides; 25(OH)D-25-hydroxyvitamin D and 1,25(OH)2D—1,25 dihydroxyvitamin D. Body variables: BMI—body mass index; A: adequate; Insuf: insufficient; Def: deficient. SAH: systemic arterial hypertension.

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
