# Peer review of "Adequacy and Vitamin D in the Preoperative Period of Roux-en-Y Gastric Bypass, Bariatric Surgery, Can Protect Metabolic Health in Metabolically Healthy and Unhealthy Individuals"

_nutrients, 2022, doi:10.3390/nu14030402_

Round 1

Reviewer 1 Report

Thank you for the opportunity to review this interesting article that considers the impact of pre-operative vitamin D levels on outcomes following bariatric surgery, mainly reported according to vitamin D 'adequacy' and by metabolic phenotype. The methods are well described and the conclusions appear to be supported by the findings. I have a few comments for improvements for ease of reading and interpretation:

Title - RYGB needs to be written in full in the title and the title needs to make it clear that it relates to bariatric surgery please (RYGB is carried out by other specialties for other reasons). 

Results - the term adequacy and inadequacy become quite confusing within the results, particularly as the terms are used to describe vitamin D levels but also adequate/inadequate insulin, cholesterol etc. The reader can refer back to previous sections to remind themselves what vitamin D level the terms refer to but the terms don't make sense when referring to the other elements. It isn't clear if an inadequacy of lipid or insulin  is a good thing or not.  Within the results i would suggest that rather than referring to adequate or inadequate vitamin D you state the cut off level as  25(O)D < or > 30ng/ml. It would also be helpful to the reader to see what the normal range for cholesterol, LDL etc is and then perhaps present results according to how patient levels deviate from the normal range pre/post operatively.

The text beneath the tables refers to statistical analysis of a number of measurements that are not described within the table, for example WC - waist circumference. If any of these are not relevant to the table they should be removed or moved to a more relevant section. Are these results that are likely to appear in a supplementary file? If not then they don't need to be mentioned within the article at all of if they are then a short paragraph to explain this is needed. 

I look forward to reading a revised version of this article

thank you

Author Response

Reviewer 1:

Thank you for the opportunity to review this interesting article that considers the impact of pre-operative vitamin D levels on outcomes following bariatric surgery, mainly reported according to vitamin D 'adequacy' and by metabolic phenotype. The methods are well described and the conclusions appear to be supported by the findings. I have a few comments for improvements for ease of reading and interpretation:

Title - RYGB needs to be written in full in the title and the title needs to make it clear that it relates to bariatric surgery please (RYGB is carried out by other specialties for other reasons). Response: The authors’ have accepted the suggestions and incorporated them in the article.

Results - the term adequacy and inadequacy become quite confusing within the results, particularly as the terms are used to describe vitamin D levels but also adequate/inadequate insulin, cholesterol etc. The reader can refer back to previous sections to remind themselves what vitamin D level the terms refer to but the terms don't make sense when referring to the other elements. It isn't clear if an inadequacy of lipid or insulin is a good thing or not. Within the results I would suggest that rather than referring to adequate or inadequate vitamin D you state the cut off level as 25(O)D < or > 30ng/ml. Response: The authors’ have accepted the suggestions and incorporated them in the article.

It would also be helpful to the reader to see what the normal range for cholesterol, LDL etc is and then perhaps present results according to how patient levels deviate from the normal range pre/post operatively. Response: In accordance with the suggestion, cut-off points have been added to the tables.

The text beneath the tables refers to statistical analysis of a number of measurements that are not described within the table, for example WC - waist circumference. If any of these are not relevant to the table they should be removed or moved to a more relevant section. Are these results that are likely to appear in a supplementary file? If not then they don't need to be mentioned within the article at all of if they are then a short paragraph to explain this is needed. Response: In accordance with the reviewer's suggestion, we have revised the manuscript and removed this information from it.

I look forward to reading a revised version of this article. Thank you.

Reviewer 2 Report

Thank you for your contributions

However, this study subjects were too small and it is hard to conclude that

25OHD adequacy may have a protective effect and metabolic benefits.

I think this changes of various metabolic markers such as HOMA-IR, cholesterol, glucose, etc

were mainly due to body weight reduction. How can you explain the better changes of metabolic markers

in 25OHD inadequacy with MUHO than MHO, even though their serum 25OHD was low at preOp and

high after OP.

This small numbers and short duration may show the good changes of metabolic markers after Op

including 25OHD change, not protective effect of serum 25OHD on other metabolic markers.

In addition, the increase of serum 25OHD was 17 => 25 ng/ml (mean), which change was not so

high, even though thier category of serum D level can be changed deficiency into insufficiency.

So, I suggest that this study conlcusion should be changed that bariatric op can influenc to have

better metabolic profies including serum 25OHD changes. This chages were more prominent in

MUHO...etc.

Author Response

Reviewer 2:

Thank you for your contributions. However, this study subjects were too small and it is hard to conclude that 25OHD adequacy may have a protective effect and metabolic benefits. Response: We have revised the conclusion and presented the results in accordance with the reviewer’s suggestions. However, it is worth mentioning that the present study had non-parametric statistical analyses in order to reduce the bias inherent to the sample size, as it has been performed in other studies with a segment similar to ours in the present study among Patients Undergoing Bariatric Surgery in Southern Brazil.

  1. Vivan, M.A.; Kops, N.L.; Fülber, E.R.; de Souza, A.C.; Fleuri, M.A.S.B.F.; Friedman, R. Prevalence of Vitamin D Depletion, and Associated Factors. Obes Sur 2019, 29, 3179-3187.
  2. dos Santos, M.T.A; Suano-Souza, F.I.; Fonseca, F.L.A; Lazaretti-Castro, M.; Sarni, R.O.S. Is There Association between Vitamin D Concentration and Body Mass Index Variation in Women Submitted to Y-Roux Surgery? J Obes 2018, 3251657.
  3. Wolf, E.; Utech, M.; Stehle, P.; Büsing, M.; Stoffel-Wagner, B.; Ellinger, S. Preoperative micronutrient status in morbidly obese patients before undergoing bariatric surgery: results of a cross-sectional study. Surg Obes Relat Dis 2015, 11, 1157-1163.

I think this changes of various metabolic markers such as HOMA-IR, cholesterol, glucose, etc were mainly due to body weight reduction. How can you explain the better changes of metabolic markers in 25OHD inadequacy with MUHO than MHO, even though their serum 25OHD was low at pre Op and high after OP. Response: In fact, a reduction in body weight can contribute to a reduction in the inflammatory process of obesity besides helping to improve some of the metabolic parameters we assessed. Additionally, weight loss increases serum 25(OH)D concentrations by reducing fat volume, with consequent release of the vitamin into circulation where the metabolite is distributed (Bruno et al., 2020). Although it was not the objective of the present study, attention was drawn to the location of body fat loss, especially visceral fat, given its relationship with metabolic changes in obesity. The proportions of fat deposit loss, whether visceral or subcutaneous, can affect serum concentrations of vitamin D metabolites since visceral fat contains approximately 20% more intact vitamin D than subcutaneous fat (Elina Hyppo¨nen, 2018). A randomized trial in subjects with abdominal obesity has suggested that a 50% reduction in visceral adipose tissue volume led to a 26% increase in serum concentrations of 25(OH)D. In addition, vitamin D, which has a protective metabolic effect, is fat-soluble. In this sense, it is “sequestered” by body fat and it is released when body weight decreases; thus, its serum concentrations may increase with the reduction of body weight. Therefore, this elevation of vitamin D may contribute to beneficial metabolic effects in patients who present some biochemical alteration when compared to those who are healthy.

This small numbers and short duration may show the good changes of metabolic markers after Op including 25OHD change, not protective effect of serum 25OHD on other metabolic markers. In addition, the increase of serum 25OHD was 17 => 25 ng/ml (mean), which change was not sohigh, even though thier category of serum D level can be changed deficiency into insufficiency. So, I suggest that this study conlcusion should be changed that bariatric op can influenc to have better metabolic profies including serum 25OHD changes. This chages were more prominent in MUHO...etc. Response: The authors’ have accepted them in the article.
